# Ethylene Acts as a Local and Systemic Signal to Mediate UV-B-Induced Nitrate Reallocation to Arabidopsis Leaves and Roots via Regulating the ERFs-NRT1.8 Signaling Module

**DOI:** 10.3390/ijms23169068

**Published:** 2022-08-13

**Authors:** Xiao-Ting Wang, Jun-Hua Xiao, Li Li, Jiang-Fan Guo, Mei-Xiang Zhang, Yu-Yan An, Jun-Min He

**Affiliations:** College of Life Sciences, Shaanxi Normal University, Xi’an 710119, China

**Keywords:** *Arabidopsis thaliana*, ERFs, ethylene biosynthesis, ethylene signaling, nitrate reallocation, NRT1.8, systemic signal, UV-B

## Abstract

Nitrate is the preferred nitrogen source for plants and plays an important role in plant growth and development. Under various soil stresses, plants reallocate nitrate to roots to promote stress tolerance through the ethylene-ethylene response factors (ERFs)-nitrate transporter (NRT) signaling module. As a light signal, ultraviolet B (UV-B) also stimulates the production of ethylene. However, whether UV-B regulates nitrate reallocation in plants via ethylene remains unknown. Here, we found that UV-B-induced expression of *ERF1B*, *ORA59*, *ERF104*, and *NRT1.8* in both Arabidopsis shoots and roots as well as nitrate reallocation from hypocotyls to leaves and roots were impaired in ethylene signaling mutants for Ethylene Insensitive2 (EIN2) and EIN3. UV-B-induced *NRT1.8* expression and nitrate reallocation to leaves and roots were also inhibited in the triple mutants for *ERF1B*, *ORA59*, and *ERF104*. Deletion of *NRT1.8* impaired UV-B-induced nitrate reallocation to both leaves and roots. Furthermore, UV-B promoted ethylene release in both shoots and roots by enhancing the gene expression and enzymatic activities of ethylene biosynthetic enzymes only in shoots. These results show that ethylene acts as a local and systemic signal to mediate UV-B-induced nitrate reallocation from Arabidopsis hypocotyls to both leaves and roots via regulating the gene expression of the ERFs-NRT1.8 signaling module.

## 1. Introduction

Nitrogen (N) is one of the basic elements of nucleic acids, hormones, amino acids and other key substances, and it is a key limitation factor for crop growth and yield. In agricultural production, application of N fertilizer increases year by year to meet the increasing food demand caused by global population growth. However, only about 30–50% of the N fertilizer invested can be absorbed and converted into agricultural productivity, and the rest is discharged into the atmosphere in the form of nitrous oxide or leached underground in the form of nitrate, causing low nitrogen use efficiency and environmental pollution [1,2]. Therefore, research on the mechanism of nitrogen uptake, transportation and distribution has gained increasing attention.

Inorganic nitrogen is the main nitrogen source for plants obtained from soil, and nitrate, based on its highly liquidity, is an ideal inorganic nitrogen source for plants [3]. Nitrate transporters (NRT) are involved in nitrate uptake, transport and distribution in plants [4]. Nitrate transportation from roots to shoots is achieved through symplast and apoplast pathways. In this process, NPF7.2/NRT1.8 and NPF7.3/NRT1.5 are two transporters mediating long-distance nitrate transport. NRT1.8, which is mainly expressed in the membrane of xylem parenchyma cells of roots, is responsible for unloading nitrate from the xylem to root cells. NRT1.5, which is mainly expressed in the pericycle cells near the protoxylem of roots, is responsible for loading nitrate into the xylem [5,6]. In addition, plants can transport nitrate upward through the phloem by NPF2.13/NRT1.7 and NPF2.9/NRT1.9 [7,8]. Once nitrate is transported to the shoots, many members of the NRT family are involved in nitrate allocation. NRT1.8 not only plays a role in unloading nitrate from the xylem to roots, but also participates in unloading nitrate from the xylem to leaves [6]. Previous studies have shown that under cadmium (Cd), salt (Na) and drought stresses, the expression of *NRT1.8* and *NRT1.5* in roots was significantly upregulated and downregulated, respectively, resulting in more nitrate accumulation in roots to improve plant stress resistance [6,9]. Further studies showed that Cd and Na stresses induced nitrate reallocation to roots via initiating ethylene/jasmonic acid (JA) signaling, which converged at ethylene insensitive 3 (EIN3)/EIN3-like1 (EIL1). EIN3/EIL1 modulates the ethylene responsive factors (ERFs) ERF104, ERF1B and ORA59 and hence upregulates *NRT1.8* expression in roots. Furthermore, EIN3/EIL1 binds to the promoter region of *NRT1.5* to inhibit its expression in roots [10]. Thus, stress-initiated nitrate allocation to roots (SINAR) is regulated by the ethylene/JA-NRT signaling module and serves as a universal mechanism of plants in response to diverse stresses [6,9,10].

Ultraviolet-B (UV-B) light, an inherent component of sunlight, is not only a potential stress factor but also a signal to regulate plant growth and development [11,12]. At present, there are UV-B-specific and -nonspecific signaling pathways in plants. The UV-B-specific signaling pathway, dependent on the UV-B receptor UV RESISTANCE LOCUS8 (UVR8), can be activated at a low dose of UV-B radiation and mainly mediates plant morphogenesis and defense gene expression [12,13,14]. The UV-B-nonspecific signaling pathway, independent of UVR8, can be activated by high-dose UV-B radiation or other stresses and mediates stress-related gene expression [15]. Furthermore, 0.5 W/m^2^ UV-B (about 3.45 µmol/m^2^/s), close to the UV-B in sunlight [16], is the biologically effective radiation [14], under which both the UV-B-specific and -nonspecific signaling pathway can be activated [17,18]. Similar to soil stress factors such as Cd, Na and drought stresses, UV-B can also induce ethylene production in many plants [19,20]. However, it remains unknown whether UV-B, similar to soil stress factors, also regulates nitrate reallocation and whether the ethylene-ERFs-NRT signaling module is also involved in this process.

Under abiotic and biotic stresses, different parts of the same plant are not exposed to the same stress intensity [21,22]. Tissues that initially sense stress signals (i.e., local tissue) send systemic signals to other tissues even to parts of the plant (i.e., systemic tissue) that have not yet been subjected to the stresses. These systemic signals induce acclimation processes in systemic tissues, termed “systemic acquired acclimation (SAA)”, enabling these tissues to prepare for the possibility of being subjected to the stresses [23]. SAA plays a vital role in optimizing plant growth and preventing damage caused by abiotic and biotic stresses [24]. Under light stress, a variety of systemic signals are activated in leaves, including electrical signals, systemic reactive oxygen species (ROS), systemic redox changes, hydraulic waves, and hormones, such as JA, abscisic acid (ABA) and auxin [24,25,26,27,28,29,30,31,32]. In the process of light-promoted root growth and nitrate uptake, bZIP transcription factor HYPOCOTYL5 (HY5) is a shoot-to-root mobile signal. Shoot-derived HY5 auto-activates root HY5 and promotes root nitrate uptake by activating *NRT2.1* [33]. Furthermore, hormone-response transcripts were identified in response to light stress [34]. However, to date, it is still unclear whether ethylene can act as a systemic signal to regulate nitrate reallocation in the systemic tissues under light stimulus or other stresses.

In this study, we provide evidence that UV-B radiation promotes gene expression and enzymatic activities of ethylene biosynthetic enzymes only in shoots, thus increasing ethylene biosynthesis in shoots (the local tissue) and promoting transportation of ethylene from shoots to roots (the systemic tissue). Ethylene in local and systemic tissues, whose function is dependent on the ethylene signaling components EIN2 and EIN3, induces the expression of genes in the ERFs-NRT1.8 signaling module, thus promoting NRT1.8-mediated nitrate unloading from hypocotyl to roots and leaves. This study not only helps to understand the signal transduction mechanism of nitrate reallocation regulated by UV-B radiation but also gives us a new insight into ethylene acting as a local and systemic signal in plants.

## 2. Results

### 2.1. UV-B Induces Nitrate Reallocation from Hypocotyls to Leaves and Roots

To investigate the effect of UV-B radiation on nitrate allocation in Arabidopsis seedlings, we determined the nitrate concentration in leaves, hypocotyls and roots of 20-day-old wild-type Col-0 seedlings under light alone or with 0.5 W/m^2^ UV-B irradiation for 3 h. Under light condition, nitrate levels in both leaves and roots were lower than that in hypocotyls (Figure 1A), leading to a leaf/hypocotyl (L/H) nitrate ratio of 0.77 and a root/hypocotyl (R/H) nitrate ratio of 0.24 (Figure 1B). UV-B irradiation significantly increased the nitrate contents in both leaves and roots but decreased that in hypocotyls (Figure 1A), resulting in an L/H ratio of 1.52 and an R/H ratio of 0.42 (Figure 1B). These results indicate that UV-B radiation promotes nitrate reallocation from hypocotyls to the local tissue leaves and the systemic tissue roots.

### 2.2. UV-B Induces Nitrate Reallocation by Promoting NRT1.8 Expression in Shoots and Roots

Considering that NRT1.8 regulates nitrate reallocation in plants under several soil stresses [6,10], we determined if NRT1.8 is involved in UV-B-regulated nitrate reallocation in Arabidopsis. We first investigated the gene expression of *NRT1.8* in shoots and roots. When seedlings grown in vermiculite substrate were exposed to 0.5 W/m^2^ UV-B radiation for 3 h, the expression of *NRT1.8* was enhanced by 14.49 times in shoots and 2.43 times in roots (Figure 2A). This result was consistent with that UV-B caused a higher nitrate accumulation in leaves than that in roots (Figure 1), suggesting the involvement of NRT1.8 in UV-B-regulated nitrate reallocation to both leaves and roots. To confirm the role of NRT1.8, we next detected the effect of *NRT1.8* mutation on UV-B-induced nitrate reallocation to leaves and roots. Under normal light condition, the nitrate levels in leaves, hypocotyls and roots were similar in WT Col-0 and *nrt1.8* (Figure 2B). The nitrate ratios of L/H and R/H in *nrt1.8* were also not significantly different from those in WT Col-0 (Figure 2C). However, under UV-B treatment, in contrast to WT Col-0, no nitrate reallocation from hypocotyls to leaves and roots was found in *nrt1.8* mutants (Figure 2B,C). This result indicates that the nitrate reallocation from hypocotyls to leaves and roots in WT Col-0 is abolished by *NRT1.8* mutation, confirming the role of NRT1.8 in mediating UV-B-induced nitrate reallocation from hypocotyls to both leaves and roots.

### 2.3. UV-B Induces Expression of ERF Genes, and ERFs Mediates UV-B-Induced NRT1.8 Expression and Subsequent Nitrate Reallocation to Leaves and Roots

Given that ERFs including ERF1B, ORA59 and ERF104 are immediate upstream regulators of *NRT1.8* [10] and that *NRT1.8* expression is greatly induced by UV-B in shoots and roots (Figure 2), we further detected whether UV-B radiation induces expression of these *ERFs* in wild-type Col-0 shoots and roots. As expected, UV-B irradiation significantly induced expression of *ERF1B*, *ORA59,* and *ERF104* in shoots and *ERF1B* and *ORA59* in roots, and the induction of these *ERFs* in shoots was even more dramatic than that in roots (Figure 3A). These results were consistent with the results that UV-B induced more dramatic *NRT1.8* expression in shoots than in roots (Figure 2A) and promoted a greater proportion of nitrate accumulation in leaves than in roots (Figure 1). To confirm the role of ERF1B, ORA59 and ERF104 in UV-B-induced *NRT1.8* expression and nitrate reallocation, *NRT1.8* expression and nitrate distribution were further compared between wild-type Col-0 and the *ora59 erf104 erf1b* triple mutants, *tri-1* and *tri-2* [10]. In the *tri-1* and *tri-2* triple mutants, UV-B-induced *NRT1.8* expression was inhibited largely in shoots and completely blocked in roots (Figure 3B). Consistent with this, UV-B-induced changes of nitrate level and nitrate ratio were all completely inhibited in *tri-1* and *tri-2* mutants (Figure 3C,D). These results firmly indicate that UV-B-induced *NRT1.8* expression and subsequent nitrate reallocation from hypocotyls to leaves, and roots are mediated by ERF transcription factors ERF1B, ORA59, and ERF104.

### 2.4. Ethylene Signaling Pathway Is Involved in UV-B-Induced Nitrate Reallocation through Regulating Gene Expression of ERFs-NRT1.8 Signaling Module

Considering that the ethylene signaling pathway regulates ERFs-NRT1.8 signaling module gene expression upon treatment with Cd, Na, and ethylene [10], we further investigated whether ethylene signaling also mediates UV-B-induced expression of ERFs-NRT1.8 module genes and the subsequent nitrate reallocation using ethylene signaling mutants. Results showed that UV-B-induced expression of *ERF* genes *ERF1B*, *ORA59*, and *ERF104* (Figure 4A) as well as *NRT* gene *NRT1.8* (Figure 4B) in both shoots and roots significantly decreased in *ein2-1* and *ein3-1* compared with wild-type Col-0. Consistent with this, *ein2-1* and *ein3-1* also significantly inhibited the rise in nitrate level in both leaves and roots and the decrease in nitrate level in hypocotyls induced by UV-B (Figure 4C). Subsequently, *ein2-1* and *ein3-1* also significantly inhibited the UV-B-increased nitrate ratios of L/H and R/H (Figure 4D). These results indicate that the ethylene signaling pathway mediates UV-B-induced nitrate reallocation from hypocotyls to both leaves and roots by regulating gene expression of the *ERFs*-*NRT1.8* signaling module. 

### 2.5. UV-B Increases Ethylene Levels in Both Shoots and Roots by Inducing Ethylene Biosynthesis Only in Shoots

Having established that ethylene signaling mediates UV-B-induced nitrate reallocation by inducing gene expression of the *ERFs*-*NRT1.8* signaling module in both shoots and roots (Figure 1, Figure 2, Figure 3 and Figure 4), we further investigated whether UV-B activates ethylene signaling by inducing ethylene biosynthesis in both shoots and roots. We first measured ethylene release rates in shoots and roots of wild-type seedlings in response to UV-B, and the results showed that UV-B radiation significantly increased ethylene release rates in both shoots and roots (Figure 5A), suggesting that UV-B induces ethylene accumulation in both shoots and roots. We further investigated UV-B’s impact on the gene expression and enzymatic activities of ethylene biosynthetic enzymes ACS and ACO in shoots and roots. When Arabidopsis seedlings grown in vermiculite substrate were exposed to 0.5 W/m^2^ UV-B for 3 h, the expression of *ACS* genes including *ACS2*, *ACS5*, and *ACS7* as well as *ACO* genes including *ACO1* and *ACO2* was induced significantly only in the UV-B-irradiated shoots but not in the UV-B-nonirradiated roots (Figure 5B,C, Appendix A). Consistent with this, UV-B treatment enhanced activities of ethylene biosynthetic enzymes ACS and ACO only in shoots but not in roots (Figure 5D). Taken together, these results not only indicate that UV-B radiation enhances ethylene biosynthesis only in the UV-B-radiated shoots and not in the UV-B-nonradiated roots, but also suggest that UV-B-induced ethylene in the local UV-B-irradiated shoots can act as a systemic signal to the UV-B-nonirradiated roots. 

## 3. Discussion

Previous studies have shown that SINAR serves as a universal mechanism of plants in response to diverse soil stresses such as salt, drought and heavy metal stresses [6,9,10], under which roots are the local tissues that initially sense the stress signals. However, whether other environmental factors, such as visible light and UV-B signals that are initially sensed by leaves (the local tissues) but not by roots (the systemic tissues), also initiate nitrate allocation to roots is still unclear. Meanwhile, it is also unclear whether light signals and soil factors can initiate nitrate allocation to the local tissue leaves and the systemic tissue leaves, respectively. In this study, our data demonstrate that UV-B radiation induces nitrate reallocation from hypocotyls of Arabidopsis seedlings not only to the local tissue leaves but also to the systemic tissue roots (Figure 1). This result further confirms that SINAR is a universal acclimation mechanism of plants in response to both soil and light stresses and suggests that UV-B can induce nitrate reallocation to both the local and systemic tissues. Whether the soil factors can also initiate nitrate allocation to the systemic tissue leaves is worth studying in the future.

Studies have shown that Cd and Na stresses induce nitrate reallocation to roots via initiating ethylene signaling to induce expression of *ORA59*, *ERF1B* and *ERF104* and hence to upregulate *NRT1.8* expression in roots [6,10]. However, stresses mentioned above are soil stresses, and roots are the directly stressed local tissues. It remains unknown whether light signal UV-B regulates nitrate reallocation to both the local tissue leaves and the systemic tissue roots also through the ethylene-ERFs-NRT1.8 signaling module. Here, we found that UV-B induced expression of *ERF104*, *ERF1B*, *ORA59*, and *NRT1.8* in both Arabidopsis shoots and roots (Figure 2 and Figure 3), suggesting that these ERFs may mediate UV-B-induced *NRT1.8* expression and the subsequent nitrate reallocation to leaves and roots. However, *ERF104* expression showed significant but less response in shoots and no response in roots to UV-B (Figure 3A). This might be due to the fact that ERF104 is preferentially regulated at the posttranslational level by UV-B-triggered signals such as ethylene [35]. Furthermore, nitrate reallocation from hypocotyls to both leaves and roots was impaired in ethylene signaling mutants for *EIN2* and *EIN3* (Figure 4). UV-B-induced *NRT1.8* expression and nitrate reallocation were also inhibited in the triple mutants for *ERF104*, *ERF1B*, and *ORA59* (Figure 3), and deletion of *NRT1.8* impaired UV-B-induced nitrate reallocation as well (Figure 2). These data not only provide evidence that UV-B induces nitrate reallocation to both the local tissue leaves and the systemic tissue roots via ethylene-ERFs-NRT1.8 signaling module, but also suggest that this signaling module mediates nitrate reallocation in plants in responses to various environmental conditions. However, our results also showed that the UV-B-induced nitrate reallocation and gene expression of *ERFs* and *NRT1.8* were only partially inhibited in Arabidopsis shoots when EIN2 and EIN3 were mutated (Figure 4). These results further suggest that except for the ethylene signaling pathway, there may be other pathways also involved in the regulation of nitrate reallocation under UV-B radiation, such as salicylic acid and jasmonic acid [10,36]. Previous studies have shown that HY5 is involved in light-regulated nitrate absorption and distribution [33], and that HY5 is also a key component of the UV-B-specific signaling pathway dependent on UVR8 [13,14,15]. Therefore, it is worth exploring whether UV-B-induced nitrate reallocation is under a coordinated regulation between ethylene- and UVR8-dependent pathways.

Under diverse abiotic and biotic stresses, the local tissues can send systemic signals to induce acclimation processes in the systemic tissues, termed SAA, which play a vital role in optimizing growth and preventing damage associated with abiotic and biotic stress conditions [24,25,26]. Studies have shown that a variety of systemic signals are activated in leaves under light stress, including electrical signals, ROS, systemic redox changes, JA, ABA, auxin, hydraulic waves and HY5 [21,22,23,24,25,26,27,28,29,30,31,32,33]. However, to date, whether ethylene can act as a systemic signal under light stimulus or other stresses is still not clear. Our results showed that UV-B radiation upregulated *ACS* and *ACO* gene expression and enhanced ACS and ACO enzymatic activities only in the local irradiated tissue shoots but increased ethylene level in both the local tissue shoots and the systemic tissue roots (Figure 5). At the same time, the suppression of the ethylene signaling pathway inhibited UV-B-induced expression of *ERFs* and *NRT1.8* as well as nitrate reallocation to both leaves and roots (Figure 4). These results further indicate that ethylene induced by UV-B in shoots acts as not only a local signal but also a systematic signal, which promotes nitrate reallocation through initiating the ERF-NRT1.8 signaling module in both the local and systemic tissues. Gaseous molecule ethylene can freely diffuse from one cell to another neighboring cell, as described in [37]. However, the long-distance transport of ethylene still needs further evidence. Ethylene’s precursor ACC can be easily transported over short and long distances, providing the plant with an elaborate system to control local and remote ethylene responses [38,39]. It has been reported that ACC is a systemic signal when the roots are exposed to stress [38]. In this study, UV-B increased ethylene levels in both shoots and roots but induced ethylene biosynthesis only in the local shoots, indicating that the enhanced ethylene in the systemic roots is either direct ethylene transport or is a result of ACC transport from shoots, which needs further study to clarify. Regardless of the form of ethylene or its precursor ACC, there is no doubt that ethylene is a potential local and systemic signal involved in UV-B-induced nitrate reallocation.

Based on our research, a potential module of the UV-B signaling pathway for regulating nitrate reallocation in *Arabidopsis thaliana* is established as follows: UV-B signaling is initiated by inducing the gene expression of *ACSs* and *ACOs* in the local tissue shoots, which results in ethylene/ACC biosynthesis in shoots and thus promotes ethylene movement to roots in the form of ethylene itself or its precursor ACC. Ethylene induces the gene expression of the ERFs-NRT1.8 signaling module in both the local tissue shoots and the systemic tissue roots and thus promotes nitrate reallocation from hypocotyls to both leaves and roots (Figure 6).

## 4. Materials and Methods

### 4.1. Plant Materials and Growth Conditions

Seeds of Arabidopsis (*Arabidopsis thaliana*) wild-type and mutant *ein2-1* (CS3071), *ein3-1* (CS8052) and *nrt1.8* (CS873532) were obtained from the Nottingham Arabidopsis Stock Centre (Nottingham, UK). Seeds of *ora59-1 erf104 erf1b* triple mutants *tri-1* and *tri-2* were kindly provided by Prof. J.-M. Gong (National Key Laboratory of Plant Molecular Genetics and National Center for Plant Gene Research, Shanghai). Mutants *ein2-1*, *ein3-1*, *nrt1.8*, *tri-1* and *tri-2* are in the Col-0 ecotype. The genotypes of all mutants were confirmed by PCR analysis [10,40]. 

Plants were grown in vermiculite substrate in a growth chamber at 22 °C with 80% relative humidity under 16 h: 8 h (light: dark) conditions with 100 µmol/m^2^/s light intensity and irrigated every 2 days with a nutrient medium containing 10 mmol/L KNO_3_, 2.5 mmol/L NH_4_NO_3_, 5 mmol/L KH_2_PO_4_, 1 mmol/L MgSO_4_, 2.5 mmol/L (NH_4_)_2_SO_4_, 0.5 mmol/L CaCl_2_, 50 µmol/L H_3_BO_3_, 12 µmol/L MnSO_4_, 1 µmol/L ZnCl_2_, 1 µmol/L CuSO_4_, 0.2 µmol/L Na_2_MoO_4_, 0.1 mmol/L Fe-EDTA. After growing for 20 days, the seedlings were selected for UV-B irradiation treatment.

### 4.2. UV-B Treatment

In this study, we chose 0.5 W/m^2^ UV-B irradiation for 3 h (equaling 5.4 kJ/m^2^ received by plants) for all UV-B treatments according to a previous study [17]. Twenty-day-old seedlings were moved into an artificial climate incubator with the same growth conditions as in the growth chamber, and UV-B irradiation was obtained for 3 h from 40 W Q-panel UV 313 lamps (Largo; its maximum output is at 313 nm) covered with 0.13 mm thick cellulose diacetate (West Design Products) to transmit radiation down to 290 nm. The desired UV-B radiation intensity (0.5 W/m^2^) was achieved by adjusting the distance between seedlings and the UV 313 lamps, measured by UV spectroradiometer (Model 742) and weighted with the generalized plant response action spectrum normalized to 300 nm.

### 4.3. RNA Extraction and qPCR Analysis

After treatments, total RNA was extracted from the shoots (aboveground parts including leaves and hypocotyls) or roots using TRIzol reagent (Invitrogen, San Diego, CA, USA) according to the manufacturer’s instructions. The total RNAs were reverse-transcribed into first-strand cDNA using HiScript^®^ⅡReverse Transcriptase (Vazyme, Nanjing, China) according to the manufacturer’s instructions. qPCR was performed with ChamQ™ SYBR qPCR Master Mix (Vazyme, Nanjing, China) on an CFX96 real-time system (Bio-Rad Laboratories, Hercules, CA, USA). The qPCR assays were performed in triplicate in a reaction volume of 20 μL with 5 μL of diluted cDNA (1:10) and SYBR Green PCR Master Mix. All amplification reactions were performed in 96-well optical reaction plates with 40 cycles of denaturation for 10 s at 95 °C, annealing for 30 s at 60 °C. For each independent biological replicate, the relative transcript level was the mean of three technical replicates. Each qPCR result was the average of three independent biological repeats. The relative expression level is shown as a value relative to that of wild-type Col-0 under light treatment after normalization to those of *ACTIN2* [41]. The primer sequences for qPCR are listed in Appendix A. Most of the primers have been successfully used in previous studies [10,40,41,42]. Amplification efficiency and specificity of the newly reported primers in the present study were checked by reading the amplification plot from a machine. The relative quantification of each gene expression was performed using the 2^−ΔΔCt^ method [43]. 

### 4.4. Determination of Nitrate Content

For nitrate content analyses, at least thirty seedlings for each biological replicate were dissected into leaves, hypocotyls and roots, and then, 3 mL of milli-Q water per 0.1 g fresh weight (FW) of tissues was added, and the mixture was boiled for 20 min and frozen at −80 °C overnight. The material was centrifuged at 20,800× *g* for 5 min at room temperature, and then, the supernatant was taken up in a 1 mL syringe, passed through a 0.22 µm filter, and its nitrate content was determined by HPLC (Agilent 1200 series) using a PARTISIL 10 strong anion-exchange column (Whatman) as described [44].

### 4.5. Detection of Ethylene Release Rate

In this study, we detected the ethylene release rate as mentioned previously [42]. After the aforementioned treatments (light or 0.5 W/m^2^ UV-B irradiation for 3 h), the shoots (aboveground parts including leaves and hypocotyls) and roots from at least thirty seedlings for each biological replicate were immediately enclosed in 2 mL vials. After 3 h, the gas in the vial was extracted using a syringe, and then, the ethylene content was measured with an Agilent 6890 NGC system equipped with a flame ionization detector on an HP-5 capillary column (Agilent Technologies, Palo Alto, CA, USA). Treatment was repeated at least three times. Data are presented as means ± standard errors (SE).

### 4.6. Assays for ACS Activity

ACS activity in shoots and roots from at least thirty seedlings for each biological replicate was detected as described previously [45] with slight modifications. Briefly, 0.1 g tissues was ground with liquid nitrogen and then resuspended in 350 μL buffer A (200 mmol/L phosphate buffer, pH 8.0, 10 μmol/L pyridoxal phosphate, 1 mmol/L EDTA, 2 mmol/L PMSF and 5 mmol/L DTT). The samples were centrifuged at 15,000× *g* for 15 min at 4 °C, and then, 300 μL extraction was transferred to a 5 mL vial containing 100 μL 5 mmol/L S-(5′-Adenosyl)-L-methionine (AdoMet). After incubation for 1 h at 22 °C, 100 μL 10 mmol/L HgCl_2_ and 100 μL 1:1 mixture of saturated NaOH:bleach were added in order to promote the conversion of ACC formed to ethylene. The reaction vials were then sealed with rubber serum stoppers and incubated on ice for 20 min. For each sample, 1 mL of headspace gas in vial was removed with a syringe and injected into gas chromatograph (Agilent 6890 NGC) for ethylene determination as described above. All reactions were performed in three replications and compared with controls, to which AdoMet was not added.

### 4.7. Assays for ACO Activity

ACO activity in shoots and roots from at least thirty seedlings for each biological replicate was assayed as described previously [46] with slight modifications. Briefly, 0.5 g tissues was ground with liquid nitrogen and then resuspended in extraction buffer (10% glycerol, 30 mmol/L sodium ascorbate, 5% polyvinyl polypyrrolidine, 0.1 mol/L Tris-HCl, pH 7.2). Homogenate of tissues was centrifuged at 15,000× *g* for 20 min at 4 °C, and then, 0.2 mL supernatant was mixed with 2 mL reaction mixture containing 1.7 mL extraction buffer (without polyvinyl polypyrrolidine), 50 μmol/L FeSO_4_, and 2 mmol/L ACC, and incubated at 30 °C. Ethylene produced in the head space of 5 mL capped tubes after a 1 h incubation was determined as described above. All reactions were performed in three replications and compared with controls, to which ACC was not added.

### 4.8. Statistical Analysis

Results from different treatments were compared using Student’s *t* test and one-way ANOVA (analysis of variance). Following ANOVA, post hoc comparisons of means were made using Mann–Whitney multiple comparisons. Statistical significance was determined at *p* < 0.05, *p* < 0.01 or *p* < 0.005, as indicated in the figure legends. All data analyses were carried out using SPSS16.0. 

## Figures and Tables

**Figure 1 ijms-23-09068-f001:**
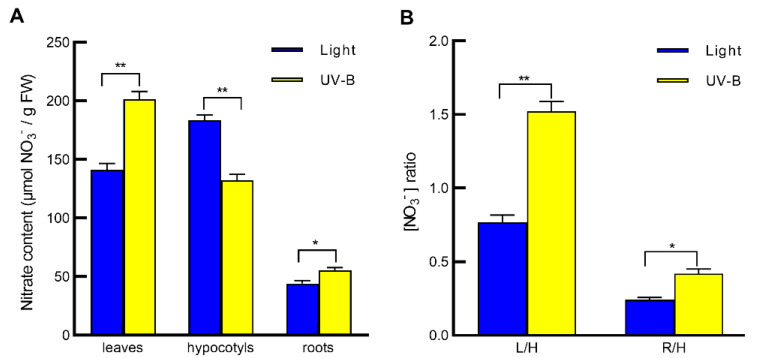
UV-B-induced nitrate reallocation from hypocotyls to leaves and roots. Twenty-day-old seedlings of Arabidopsis wild-type Col-0 grown in vermiculite substrate were exposed to light alone (Light) or with 0.5 W/m^2^ UV-B (UV-B) for 3 h. Then, leaves, hypocotyls and roots were harvested, their nitrate contents were determined (**A**), and the leaf/hypocotyl (L/H) and root/hypocotyl (R/H) nitrate ratios were calculated (**B**). Values are means ± SE of three biological replicates, each pooled more than thirty plants. Asterisks indicate significant differences between light and UV-B treatments (* *p* < 0.05; ** *p* < 0.01).

**Figure 2 ijms-23-09068-f002:**
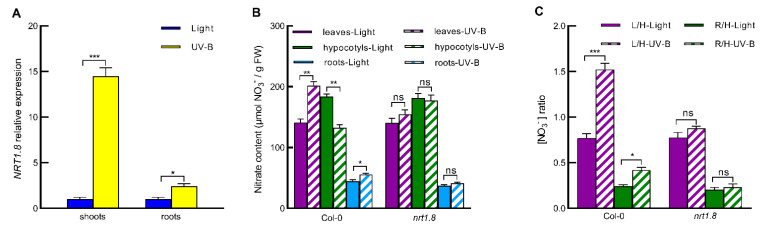
UV-B induced *NRT1.8* expression and NRT1.8 mediated UV-B-induced nitrate reallocation to leaves and roots. (**A**) *NRT1.8* expression was determined by qPCR in shoots and roots of wild-type Col-0 exposed to light alone (Light) or with 0.5 W/m^2^ UV-B (UV-B) for 3 h. Data are presented as values relative to those of shoots and roots under light treatments after normalization to those of *ACTIN2*. (**B**) Nitrate content was determined in leaves, hypocotyls and roots of wild-type Col-0 and *nrt1.8* plants exposed to light alone (Light) or with UV-B (UV-B) for 3 h. (**C**) Leaf/hypocotyl (L/H) and root/hypocotyl (R/H) nitrate ratios were calculated after determination of nitrate levels. Values are means ± SE (*n* = 3). Asterisks indicate significant differences between light and UV-B treatments (* *p* < 0.05; ** *p* < 0.01; *** *p* < 0.005; ns: no significant difference).

**Figure 3 ijms-23-09068-f003:**
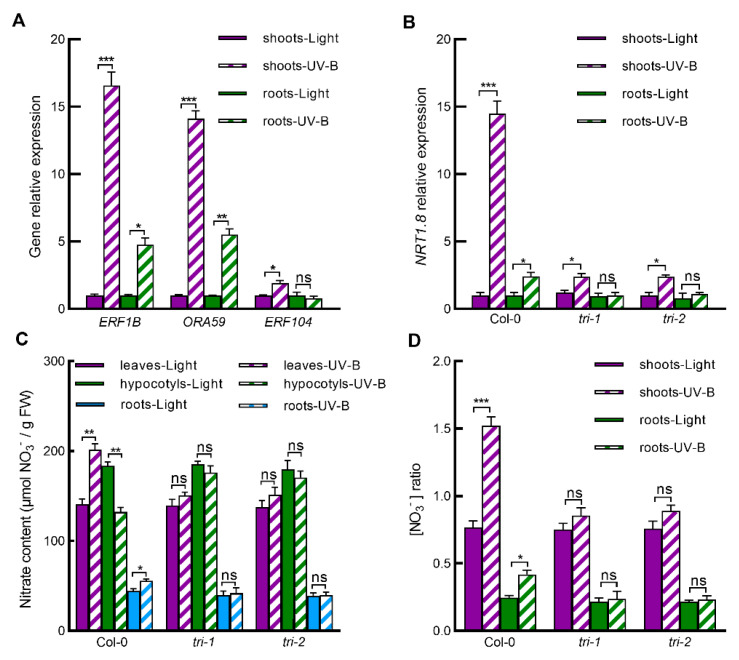
UV-B induced expression of *ERF* genes, and ERFs mediated UV-B-induced *NRT1.8* expression and nitrate reallocation to leaves and roots. (**A**,**B**) Expression of *ERF1B*, *ORA59*, and *ERF104* in shoots and roots of wild-type Col-0 (**A**) and expression of *NRT1.8* in shoots and roots of wild-type Col-0 and the *ora59 erf104 erf1b* triple mutants *tri-1* and *tri-2* (**B**) were determined by qPCR in plants exposed to light alone (Light) or with 0.5 W/m^2^ UV-B (UV-B) for 3 h. Data are presented as values relative to those under light treatments after normalization to those of *ACTIN2*. (**C**,**D**) Nitrate concentration was determined in leaves, hypocotyls and roots of wild-type Col-0, *tri-1* and *tri-2* plants exposed to light alone (Light) or with UV-B (UV-B) for 3 h (**C**), and then leaf/hypocotyl (L/H) and root/hypocotyl (R/H) nitrate ratios were calculated (**D**). Values are means ± SE (*n* = 3). Data with asterisks indicate significant differences between light and UV-B treatments (* *p* < 0.05; ** *p* < 0.01; *** *p* < 0.005; ns: no significant difference).

**Figure 4 ijms-23-09068-f004:**
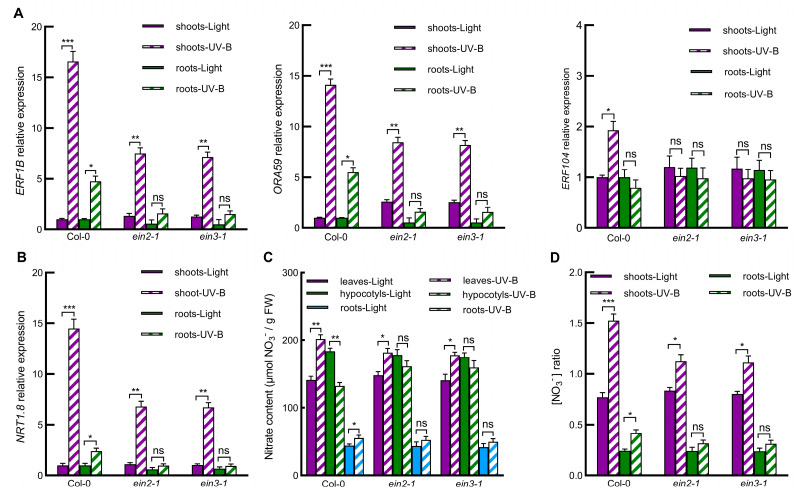
The ethylene signaling pathway was involved in UV-B-induced nitrate reallocation through regulating gene expression of the *ERFs*-*NRT1.8* signaling module. (**A**,**B**) Expression of *ERF1B*, *ORA59*, and *ERF104* (**A**) or *NRT1.8* (**B**) was determined by qPCR in shoots and roots of wild-type Col-0, *ein2-1*, and *ein3-1* plants exposed to light alone (Light) or with 0.5 W/m^2^ UV-B (UV-B) for 3 h. Data are presented as values relative to that of wild-type Col-0 under light treatment after normalization to those of *ACTIN2*. (**C**,**D**) Nitrate concentration was determined in leaves, hypocotyls and roots of wild-type Col-0, *ein2-1* and *ein3-1* plants exposed to light alone (Light) or with UV-B (UV-B) for 3 h (**C**), and then leaf/hypocotyl (L/H) and root/hypocotyl (R/H) nitrate ratios were calculated (**D**). Values are means ± SE (*n* = 3). Values with asterisks indicate significant differences between light and UV-B treatments (* *p* < 0.05; ** *p* < 0.01; *** *p* < 0.005; ns: no significant difference).

**Figure 5 ijms-23-09068-f005:**
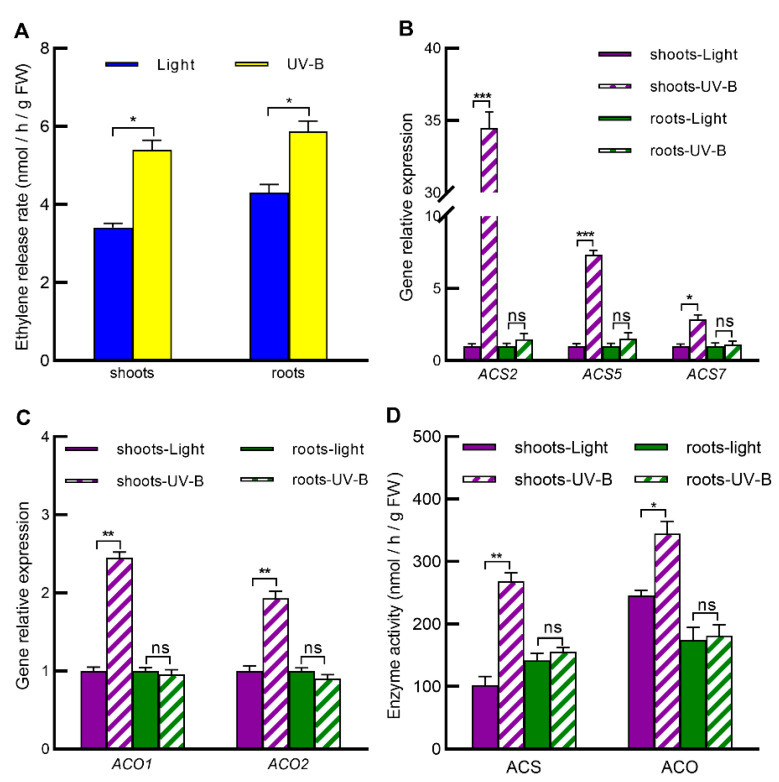
UV-B enhanced ethylene levels in both shoots and roots by only increasing ethylene biosynthesis in Arabidopsis shoots. Seedlings of Arabidopsis wild-type Col-0 grown in vermiculite substrate were exposed to light alone (Light) or with 0.5 W/m^2^ UV-B (UV-B) for 3 h. Then, shoots and roots were harvested and subjected to determination of ethylene release rates (**A**), gene expression of *ACS* (**B**) and *ACO* (**C**), and activities (**D**) of ethylene biosynthetic enzymes ACS and ACO. Values are means ± SE of three biological replicates, each pooled more than thirty plants. Data of gene relative expression are presented as values relative to those under light treatment after normalization to those of *ACTIN2*. Data with asterisks indicate significant differences between light and UV-B treatments (* *p* < 0.05; ** *p* < 0.01; *** *p* < 0.005; ns: no significant difference).

**Figure 6 ijms-23-09068-f006:**
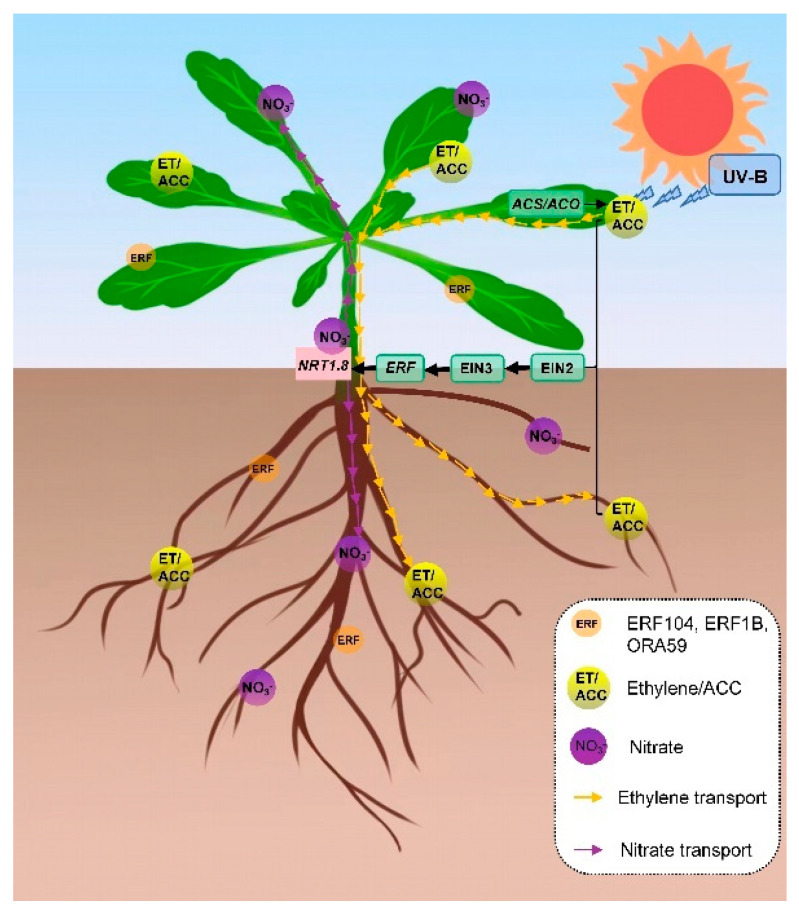
Potential module of the UV-B signaling pathway for regulating nitrate reallocation from hypocotyls to leaves and roots. UV-B signaling is initiated by inducing gene expression of *ACSs* and *ACOs* in shoots, which results in ethylene/ACC biosynthesis in shoots and thus promotes ethylene/ACC movement from shoots to roots. Ethylene activates the ERFs-NRT1.8 signaling module in both shoots and roots via EIN2- and EIN3-dependent manners and thus promotes nitrate reallocation from hypocotyls to leaves and roots.

## Data Availability

Data will be made available upon request.

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
