# Peer review of "Ethylene Acts as a Local and Systemic Signal to Mediate UV-B-Induced Nitrate Reallocation to Arabidopsis Leaves and Roots via Regulating the ERFs-NRT1.8 Signaling Module"

_ijms, 2022, doi:10.3390/ijms23169068_

Round 1

Reviewer 1 Report

I enjoyed reading the manuscript entitled "Ethylene acts as a local and systematic signal to mediate UV-B induced nitrate allocation to Arabidopsis leaves and roots via regulating ERFs-NRT1.8 signaling module". I have few comments about the research design.

1. What is the basis of UV-B dose? Why the authors chose 0.5 w/m2 UV-B? is there any reason for that? 0.5w/m2 is low dose or high dose of UV-B? In 3 hrs how much UV-B dose can plants get? Can authors put UV-B dose in KJm2d-1.

2.  In all the statistical analysis, the authors did not mentioned which test they used for the analysis? 

3. In Figure2, line no 133, the authors mentioned UV-B stress for 3 hours but did not mentioned the actual dose? is this also 0.5W/m2?

4.  The authors use qPCR as main experiment to support their hypothesis, so correct analysis of qPCR is very important. In my experience, qPCR is very tricky. How authors chose housekeeping genes such as ACTIN2? Is the authors use some other house keeping genes or only ACTIN2 as internal control? The authors should use some other house keeping genes to compare the results to make sure, internal standard are correct. Secondly, what method was used to analyzed the relative gene expression? Authors should mentioned clearly in the method section. Also, is the authors check primer/PCR efficiency? Since qPCR is the main evidence so authors should provide all the raw qPCR data as supplement files. What are the controls used to check the qPCR data accuracy? 

5.  The authors use mutants such as ein2-1, ein3-1. Are they knockout mutant? The authors should provide reverse transcriptase/qPCR data as evidence.  It would be nice to see either these single and triple mutant are knockout or knockdown mutant.

Some minor comments:

6. The authors should only discuss their results under the result section.  The line number 165-66, should move to discussion section.

7. Re write the line 37-39. The message is not clear.

Author Response

Thanks for your positive comments and detailed suggestions. We have carefully modified the whole manuscript including the language according to your and other reviewers’ suggestions. We have highlighted all the changes using the “Track Changes” function in the new manuscript. The detailed corrections are listed point by point. Please see the attachment.

Reviewer 2 Report

Title: «Ethylene acts as a local and systemic signal to mediate UV-B-induced nitrate reallocation to Arabidopsis leaves and roots via regulating ERFs-NRT1.8 signaling module».

Authors: Xiao-Ting Wang, Jun-Hua Xiao, Li Li, Jiang-Fan Guo, Mei-Xiang Zhang, Yu-Yan An*, Jun-Min He*.

Undoubtedly, the redistribution of nitrate in plants under stressful conditions plays an important role in their acclimation and resistance. In many cases, the action of stress factors develops with the participation of the ethylene signaling pathway, starting with the activation of the synthesis of this phytohormone.

In this article, an experimental attempt is made to show the local and systemic signaling role of ethylene in UV-B-induced nitrate reallocation. Works of this kind are very important, but sometimes methodologically difficult.

In this regard, I will allow here to express the main thoughts about this manuscript.

1. The article presents evidence that UV-B increases ethylene levels in both shoots and roots. This is not to say that this is a significant increase, but it is. However, a clear methodological flaw should be noted. After exposure to UV-B (it is not clear how long after exposure), the authors separate the shoots (aerial parts, including leaves and hypocotyl) and roots of at least thirty seedlings and place in a vial (unspecified volume) and hermetically seal. Ethylene is measured after 3 hours.

a) Mechanical damage (separation of the shoot from the root) to plant tissues causes a rapid and powerful burst of ethylene release, often overriding the effect of any other impact.

There are a lot of publications on ethylene production after plant wounding, including differential expression of ACS genes. For example:

Konze, J. R., & Kwiatkowski, G. M. (1981). Rapidly induced ethylene formation after wounding is controlled by the regulation of 1-aminocyclopropane-1-carboxylic acid synthesis. Planta, 151(4), 327-330.

Peck, S. C., & Kende, H. (1998). Differential regulation of genes encoding 1-aminocyclopropane-1-carboxylate (ACC) synthase in etiolated pea seedlings: effects of indole-3-acetic acid, wounding, and ethylene. Plant molecular biology, 38(6), 977-982.

Watanabe, T., Seo, S., & Sakai, S. (2001). Wound-induced expression of a gene for 1-aminocyclopropane-1-carboxylate synthase and ethylene production are regulated by both reactive oxygen species and jasmonic acid in Cucurbita maxima. Plant Physiology and Biochemistry, 39(2), 121-127.

Wang, K. L. C., Li, H., & Ecker, J. R. (2002). Ethylene biosynthesis and signaling networks. The plant cell, 14(suppl_1), S131-S151.

b) The maximum release of ethylene is usually observed several hours after UV-B irradiation. The article does not indicate how long after UV-B irradiation ethylene was determined. Perhaps, having data on the dynamics of ethylene release from separated parts of plants in the control variant and after UV-B irradiation, it would be possible to find a compromise time frame for determining the effect of UV-B on ethylene production.

2. It is considered good form in publications on the effect of UV-B on plants to indicate not only the intensity of the radiation source (W/m2), but the dose received by the plants (kJ/m2).

3. The article does not state how long after exposure to UV-B, nitrate analyzes and quantitative PCR were performed. This is important for understanding the time frame of the events taking place.

4. Mutants, in particular ein2-1, were used in the work, and it is indicated that the genotypes of all mutants were confirmed by PCR analysis. It is highly desirable to provide primer sequences for genotyping (and additional methodological information if needed). This is especially important because UV-B and hence ethylene upregulate ERF1B and ORA59 expression in ethylene signal mutants.

5. In the caption to Fig. 4 (C and D) incorrectly labeled mutants (instead of tri-1 and tri-2, ein2-1 and ein3-1 should be used).

Finally, the most important.

The article discusses the «ethylene movement from shoots to roots», which is reflected in fig. 6. This is based on indirect assumptions. However, this conclusion is most likely wrong. It is not ethylene that can move through the plant, but its precursor ACC in the form of conjugates: 1-malonyl-ACC, γ-glutamyl-ACC and jasmonyl-ACC.

The ACC received by the roots will easily turn into ethylene and start the signaling pathway. There is no need for additional expression of ACC oxidases to convert the ACC additionally received from the shoots; the existing active ACC oxidases will also cope.

Van de Poel, B., & Van Der Straeten, D. (2014). 1-aminocyclopropane-1-carboxylic acid (ACC) in plants: more than just the precursor of ethylene! Frontiers in plant science, 5, 640.

Vanderstraeten, L., & Van Der Straeten, D. (2017). Accumulation and transport of 1-aminocyclopropane-1-carboxylic acid (ACC) in plants: current status, considerations for future research and agronomic applications. Frontiers in plant science, 8, 38.

Of course, the manuscript needs to be revised and rethought.

Author Response

(The authors gave the same response as above.)

Reviewer 3 Report

Investigation of the systemic effects of ethylene is an interesting and hot topic in current plant biology. The manuscript is based on up-to-date investigations but some questions remained unanswered:

1. Authors showed that NO3- was reallocated from the hypocotyl (Fig. 1) but did not investigate the other changes in hypocotyls such as NRT1.8 expression, ET levels, and ET metabolism.

2. It was not shown how changes in nitrate content and ET level in uvr8 and hy5 mutants.

3. Selected ERFs are induced by SA and JA (Bethke et al. 2009) which could be important in the aspect of the discussion of the results.

4. How can systemically act ET from the leaves below the ground in roots?

5. How changed ROS and especially nitric oxide levels if GSNO can be systemically transported?

6. What is the impact of the reallocated NO3- on ET in the roots?

7. How changed ET level and ERFs transcripts in the roots of nrt1.8 mutants?

Author Response

(The authors gave the same response as above.)

Round 2

Reviewer 2 Report

In general, I am almost satisfied with the revision of the manuscript.

You can leave it to the readers.

It is important that the authors discuss the role of ACC in the systemic response of plants. As for the transport of ethylene over long distances, this is hardly the case. Ethylene will run out of the plant before it reaches the roots. It is necessary (in future) to carefully look at the old works with isotopically labeled ethylene.

I would like the authors to carefully look at which articles they refer to.

For example, «In this study, we detected ethylene release rate as mentioned previously [39]». But [39] is not about that!

 Please give the correct names and surnames of the authors in references [37] and [38] (use the original articles).

Also, on line 358, you need to write kJ/m2 and not KJ/m2.

Reviewer 3 Report

The authors responded to all of my questions.
